# Valence-encoding in the lateral habenula arises from the entopeduncular region

**Hao Li, Dominika Pullmann, Thomas C Jhou***

Department of Neuroscience, Medical University of South Carolina, Charleston, United States

**Abstract** Lateral habenula (LHb) neurons are activated by negative motivational stimuli and play key roles in the pathophysiology of depression. Prior reports suggested that rostral entopeduncular nucleus (rEPN) neurons drive these responses in the LHb and rostromedial tegmental nucleus (RMTg), but these influences remain untested. Using rabies viral tracers, we demonstrate disynaptic projections from the rEPN to RMTg, but not VTA, via the LHb in rats. Using in vivo electrophysiology, we find that rEPN or LHb subpopulations exhibit activation/inhibition patterns after negative/positive motivational stimuli, similar to the RMTg, while temporary inactivation of a region centered on the rEPN decreases LHb basal and burst firing, and reduces valence-related signals in LHb neurons. Additionally, excitotoxic rEPN lesions partly diminish footshock-induced cFos in the LHb and RMTg. Together, our findings indicate an important role of the rEPN, and possibly immediately adjacent hypothalamus, in driving basal activities and valence processing in LHb and RMTg neurons.
DOI: https://doi.org/10.7554/eLife.41223.001

## Introduction

The lateral habenula (LHb) has been implicated in processing aversive events across many species (*Jhou et al., 2013*; *Matsumoto and Hikosaka, 2008*; *Salas et al., 2010*; *Stopper and Floresco, 2014*; *Wang et al., 2017*). Neurons in the LHb are activated by negative motivational stimuli, which in turn suppress dopamine (DA) firing via activation of GABAergic neurons in the rostromedial tegmental nucleus (RMTg) (*Brown et al., 2017*; *Hong et al., 2011*; *Jhou et al., 2009a*; *Jhou et al., 2009b*; *Ji and Shepard, 2007*; *Smith et al., 2019*; *Vento et al., 2017*). Dysregulation of this pathway can lead to cognitive disorders associated with abnormal processing of motivational stimuli including depression (*Baker et al., 2016*; *Elmer et al., 2019*; *Lawson et al., 2017*; *Shumake and Gonzalez-Lima, 2003*). Furthermore, animal models of depression are accompanied by synaptic potentiation onto LHb neurons in mice, which can be alleviated with antidepressants (*Li et al., 2011*; *Shabel et al., 2014*), suggesting an important role of LHb afferents in mediating depressive symptoms. However, the identity of such afferents is unknown, as LHb neurons receive inputs from many brain regions (*Yetnikoff et al., 2015*), and the sources driving LHb basal activities and responses to motivational stimuli remain incompletely characterized.

Studies in mice have indicated a close anatomical and functional relationship between LHb neurons and neurons in the rostral entopeduncular nucleus (rEPN), a region homologous to the internal globus pallidus (GPi) in primates, and sometimes referred to as the habenular-projecting globus pallidus (GPh) (*Stephenson-Jones et al., 2016*). For example, the rEPN projections to the LHb can be excitatory and aversive in rodents, while primate GPi neurons respond to negative stimuli similarly to LHb neurons (*Bromberg-Martin et al., 2010*; *Hong and Hikosaka, 2008*; *Rajakumar et al., 1994*; *Shabel et al., 2012*; *Stephenson-Jones et al., 2016*). These data suggest that rEPN neurons would drive LHb responses to motivational stimuli, but such an influence has not been directly tested. In fact, LHb neurons also receive inputs from several other regions which are also involved in

*****For correspondence:**
jhou@musc.edu

**Competing interests:** The authors declare that no competing interests exist.

motivational processing (*Lecca et al., 2017*; *Root et al., 2014*; *Tooley et al., 2018*), raising further questions about the relative role of the rEPN versus other LHb afferents in valence encoding.

In the present study, we compared shock-induced neural activity in several LHb-projecting regions using cFos expression and retrograde tracing from the LHb. We further characterized response patterns of rEPN and LHb neurons to both reward and footshocks, as well as cues predicting these stimuli using in vivo electrophysiology. Finally, we examined the effects of rEPN inactivation on LHb basal activities and responses to these stimuli, and tested effects of rEPN lesions on footshock-induced cFos in the LHb and RMTg. Taken together, our findings suggest a surprising complexity of both LHb firing patterns and the selectivity of rEPN influences on them.

## Results

### LHb-projecting neurons in the rEPN and the LH are significantly activated by repeated footshocks

In order to identify LHb afferents that respond to aversive stimuli, we first used cFos as a proxy of neural activity and labeled LHb-projecting neurons by injecting the retrograde tracer cholera toxin subunit B (CTb) into the LHb (*Figure 1A*). On the test day, animals were either given a series of footshocks (30 shocks, 0.7mA each), or remained undisturbed in their home cage, before being sacrificed 1 hr later (n = 7 per group). We calculated the proportion of CTb-positive neurons expressing cFos in several LHb afferents including the ventral pallidum (VP), lateral preoptic area (LPO), rEPN, lateral hypothalamus (LH), and ventral tegmental area (VTA). We found that footshocks excited over 80% of LHb-projecting neurons in the rEPN (p<0.0001, repeated measures two-way ANOVA, Holm-Sidak correction for multiple comparisons in this and all subsequent ANOVAs) (*Figure 1B,C*), while unshocked controls exhibited markedly less cFos in these neurons. Footshock also increased cFos in LHb-projecting neurons in the LH, which interestingly were located in the ventrolateral portion of the lateral hypothalamus (vlLH), adjacent to the rEPN, but not in the dorsolateral portion (dlLH) (p<0.0001 and p=0.127 for vlLH and dlLH, repeated measures two-way ANOVA) (*Figure 1D–E*). However, the proportion of CTb-labeled cells expressing cFos in the vlLH was much less than the rEPN (p<0.0001, repeated measures two-way ANOVA). We did not see shock-induced cFos increases in CTb-labeled cells in the other regions examined (*Figure 1G*). Finally, we observed that not only were most LHb-projecting neurons in the rEPN excited by footshocks, but conversely 97% of cFos-expressing neurons in the rEPN after footshock were in turn LHb-projecting, again with this percentage higher than in the other regions examined (p=0.01 and p<0.0001 compared to the vlLH and to the remaining regions, respectively, one-way ANOVA) (*Figure 1H*). rEPN neurons innervate LHb neurons projecting to the RMTg.

To confirm serial connectivity between shock-activated LHb afferents and LHb projections to the midbrain, we used rabies virus to trans-synaptically label afferents innervating LHb neurons that either project to the RMTg or VTA. Specifically, we injected retrogradely transported virus expressing Cre-recombinase (Cav2-cre) into the RMTg or VTA and virus expressing flox-stopped an avian tumor virus receptor (TVA) and rabies glycoprotein (RG) into the LHb in six rats (n = 3 per group) (*Figure 2A*, *Figure 2—figure supplement 1A*). After waiting 3 weeks to allow TVA and RG protein to be expressed in LHb neurons that projected to either the RMTg or VTA, we injected EnvA-ΔG-rabies virus into the LHb (*Figure 2B*). As inputs to the RMTg and the VTA arise from separate LHb subpopulations (*Li et al., 2011*), this allowed us to label the regions projecting to either subset of LHb neurons (RMTg- or VTA-projecting) (*Wickersham et al., 2007*). When animals were sacrificed 1 week later, we tabulated expression of TVA protein and first-order rabies-infected cells in the LHb, along with second-order infected cells projecting to either subset of LHb neurons (*Figure 2D*, *Figure 2—figure supplement 1B*). To quantify these inputs, we calculated the proportion of inputs for each region (ratio of second-order infected cells in each region over total infected cells). Both subsets of LHb neurons received inputs from the VP, LPO, and LH, while the rEPN and VTA preferentially projected to RMTg-projecting LHb neurons (p=0.042 for both the rEPN and the VTA, two-way ANOVA) (*Figure 2C,E*, *Figure 2—figure supplement 1C-F*).

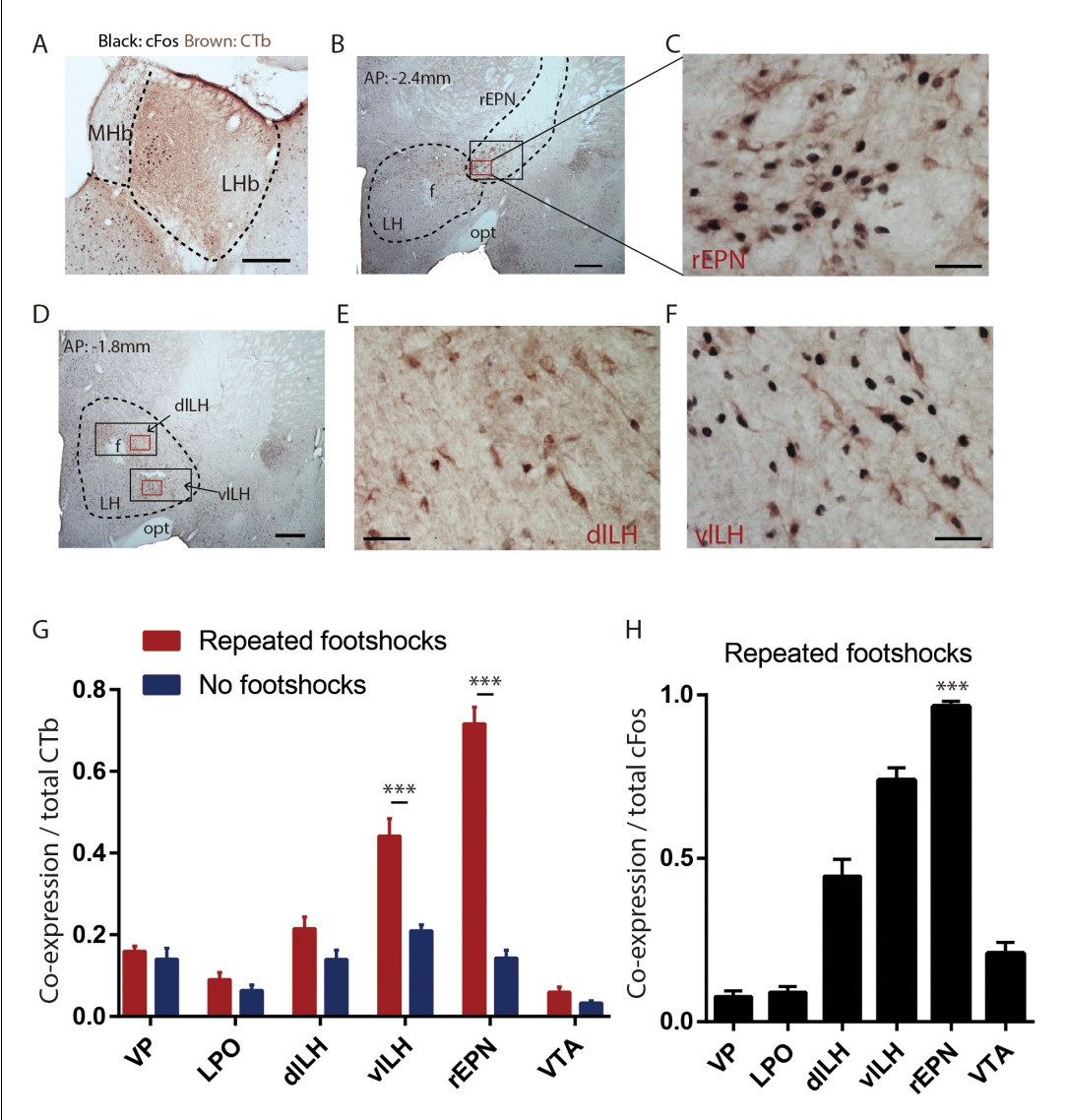

**Figure 1.** LHb-projecting neurons in the rEPN and the LH are significantly activated by repeated footshocks. (**A**) Representative photos of CTb injection site in LHb (brown immunoreactive product). Representative photos of CTb immunostaining (brown) and shock-induced cFos (black) in the rEPN (**B, C**) and the LH (**D–F**). Black squares: areas for cFos counting. Red squares: areas enlarged in representative images. f: fornix; opt: optic tract; MHb: medial habenula. (**G**) Proportion of CTb-labeled neurons expressing cFos in the VTA, rEPN, dlLH, vlLH, LPO, and VP, showing the largest shock-induced increases in the rEPN and LH. (**E**) Proportion of cFos neurons expressing CTb in these same regions. Scalebars 250 μm (**A**), 500 μm (**B, D**), 100 μm (**C, E and F**).

DOI: https://doi.org/10.7554/eLife.41223.002

## Reward cue-inhibited rEPN neurons encode valence

Given the large proportion of neurons in the rEPN activated by footshock and the connection between rEPN neurons and the RMTg-projecting LHb neurons, we next examined rEPN electrophysiological responses to motivational stimuli in a Pavlovian conditioning paradigm. Animals were pretrained with three distinct auditory tones (lasting for 2 s) predicting either a sucrose pellet, footshock (0.6 mA, 10–30 ms duration), or nothing (*Figure 3A,B*). In recording sessions, reward and shock deliveries were omitted randomly in 20% of trials, to examine rEPN responses to prediction errors (*Figure 3D*). Within 2 s of cue onset, animals approached the food port in 78% of reward trials but only 19% and 21% of shock and neutral trials, demonstrating accurate discrimination (*Figure 3D*). Among 134 recorded neurons, 53 were localized to the rEPN as identified by a lack of parvalbumin

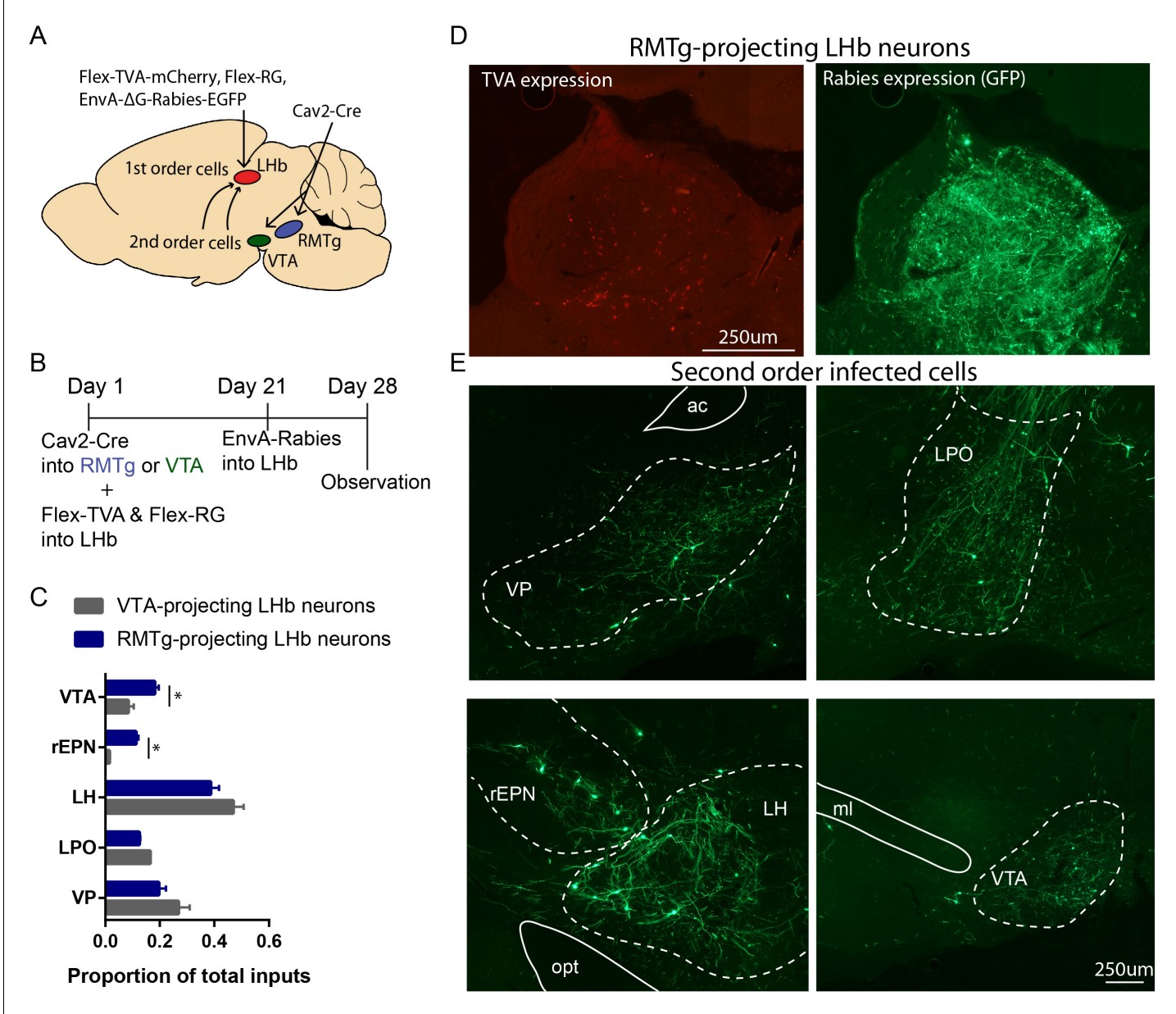

**Figure 2.** rEPN neurons innervate LHb neurons projecting to the RMTg. (**A, B**) Schematics of procedure in which retrogradely transported virus expressing Cre-recombinase was injected into the RMTg or VTA and viruses expressing flox-stopped TVA and RG were injected into the LHb. EnvA-Δ G-rabies virus was injected into the LHb 3 weeks later. Animals were perfused at the end of the forth week. (**C**) Neurons in the rEPN and the VTA preferentially projected to RMTg-projecting LHb neurons over VTA-projecting LHb neurons. (**D**) Representative pictures of RMTg-projecting TVA expressing cells and first-order rabies infected cells in the LHb. (**E**) Representative pictures of second-order rabies infected cells in the VP, LPO, rEPN, LH, and VTA after Cav2-Cre injection into RMTg, indicating neurons disynaptically connected to RMTg.

DOI: https://doi.org/10.7554/eLife.41223.003

The following figure supplement is available for figure 2:

**Figure supplement 1.** Representative images of neurons innervating VTA-projecting LHb neurons.

DOI: https://doi.org/10.7554/eLife.41223.004

immunostaining (*Rajakumar et al., 1994*), and presence of a high density of retrogradely labeled neurons after injections of the retrograde tracer CTb into the LHb (*Figure 3C*). Hence, although we did not conclusively identify the projection targets of our recorded neurons, a high proportion are likely to be LHb-projecting. Consistent with previous studies showing that LHb-projecting EPN

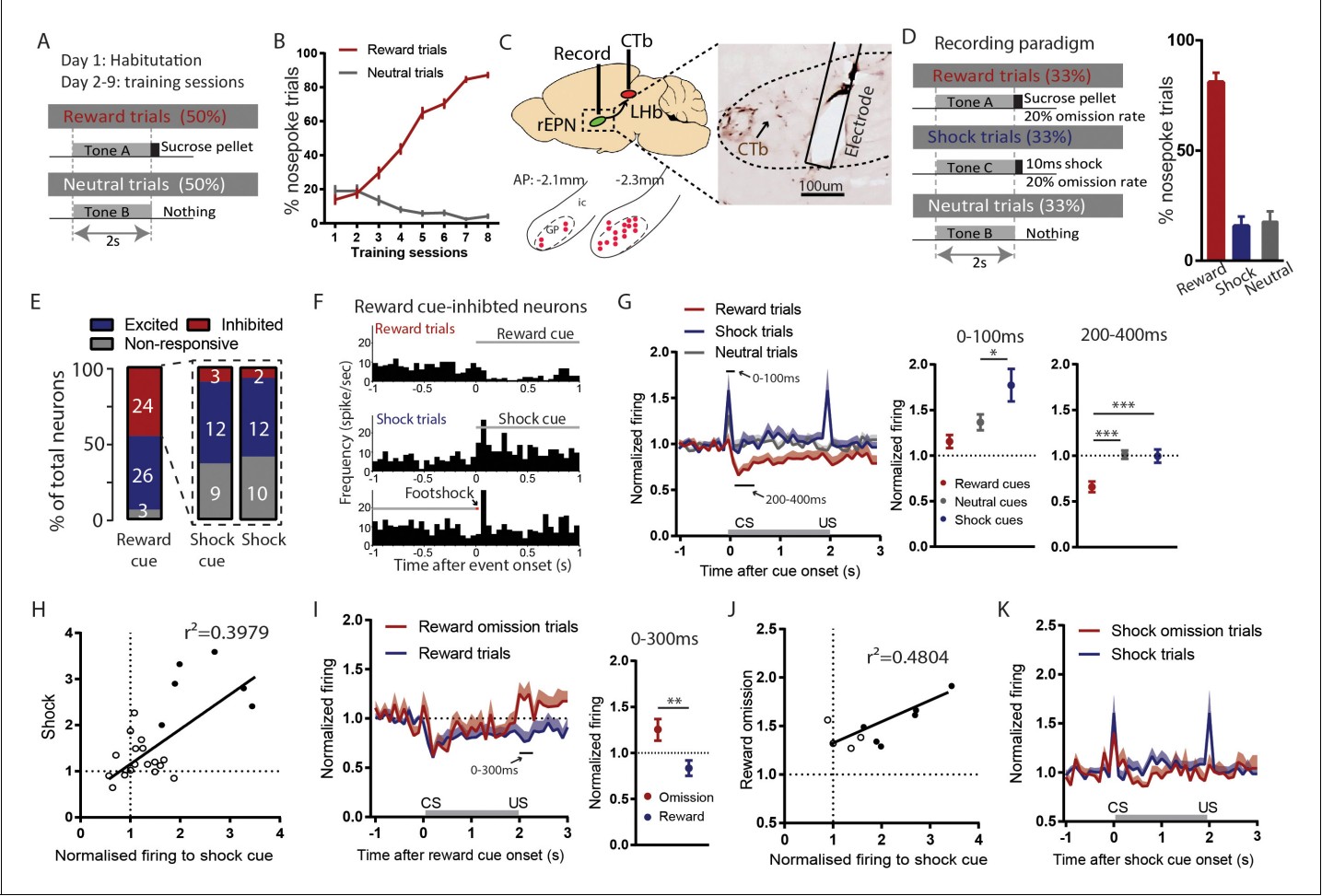

**Figure 3.** Reward cue-inhibited rEPN neurons encode valence. (A) Schematic of training paradigm. (B) After eight training sessions, rats show clear discrimination between reward and neutral cues, as measured by the probability of animals making nose pokes during the 2 s cue presentation. (C) We recorded in the rEPN, as delineated by retrograde labeling of CTb from the LHb and absence of parvalbumin immunoreactivity. (D) Schematic of recording paradigm, in which three different auditory tones signaled delivery of sucrose pellets, mild footshocks, or nothing. Animals exhibited ~80% accuracy in discrimination between different cues. (E) rEPN neurons were divided roughly equally between reward cue-inhibited and –excited responses. Amongst reward cue-inhibited neurons, the most common response was excitation to footshocks and shock cues. (F) Representative histograms of individual rEPN neuron responses to reward and shock trials, analyzed in 50 ms bins. (G) Responses of reward cue-inhibited subpopulation to reward, shock, and neutral trials, again in 50 ms bins. Excitation to shock cue was most prominent in first two bins (0–100 ms, upper solid black line), while inhibition to reward cue was most prominent 200–400 ms after stimulus onset (lower solid black line). Average firing rates from these two windows are shown in adjacent graphs. (H) In reward cue-inhibited rEPN neurons, responses to shock cues were positively correlated with individual neuron's responses to footshocks. (I) Reward cue-inhibited neurons also showed excitation to reward omission (0–200 ms window) (J) Responses to reward omissions were positively correlated with responses to shock cues in omission-activated neurons. (K) Reward cue-inhibited neurons did not respond to shock omissions. Solid circles indicate neurons significantly excited by both stimuli.

DOI: https://doi.org/10.7554/eLife.41223.005

The following figure supplement is available for figure 3:

**Figure supplement 1.** rEPN neurons not inhibited by reward cues do not show valence-like encoding patterns.
DOI: https://doi.org/10.7554/eLife.41223.006

neurons exhibit significant inhibition to reward cues (*Hong and Hikosaka, 2008*; *Stephenson-Jones et al., 2016*), we observed that a high proportion of our recorded rEPN neurons also showed inhibition by reward cues 200–400 ms post-stimulus (24/53 recorded neurons) (p<0.05, Mann-Whitney U test) (*Figure 3E,F*). Hence, we focused our subsequent analyses on these reward cue-inhibited neurons.

The population average of reward-cue inhibited rEPN neurons showed significant excitations to shock cues and footshocks during 0–100 ms post-stimulus time window, when these responses were most prominent (p=0.01 and p=0.007, respectively). These neurons were also excited by neutral cues (p=0.047), but to a lesser degree than to shock cues (p=0.0037, repeated measures one-way ANOVA, Holm-Sidak correction for multiple comparisons) (*Figure 3G*), consistent with a valence-encoding pattern, that is opposite directions of responses to positive and negative motivational stimuli. When individual neuron responses were analyzed, we found that more than half of reward cue-inhibited neurons showed significant excitation to either shock cues (12/24) or footshocks (12/24) during a 0–100 ms post-stimulus window, again consistent with valence-encoding. A smaller proportion showed no response to either shock cues (9/24) or footshocks (10/24), while the smallest proportion showed significant inhibition (3/24 and 2/24 for shock cues and footshocks) (p<0.05, Mann-Whitney U test) (*Figure 3E*). Interestingly, only six neurons were excited by both shocks and shock cues, while six neurons each were excited by either shocks and shock cues but not both, although the magnitudes of responses to shocks and shock cues were ultimately correlated with each other ($r^2$ = 0.3979, p=0.001) (*Figure 3H*). Additionally, average responses to reward omission in these reward-cue inhibited neurons were significantly greater than responses to reward during the period 200–400 ms post-stimulus (p=0.004, paired t-test) (*Figure 3I*), with 10/24 being significantly activated, and having magnitudes positively correlated with their responses to shock cues ($r^2$ = 0.4804, p=0.026) (*Figure 3J*). However, we did not observe responses to shock omissions in the population average (*Figure 3K*), and only 4/25 neurons significantly were inhibited by shock omission, a proportion not significantly different from chance (p=0.185, Chi-square).

Among EPN neurons that did *not* exhibit inhibition to reward cues (n = 29), 26/29 neurons showed excitations to reward cues. Excitations were either 'phasic', and confined to a period 100 ms post stimulus (15/29) (*Figure 3—figure supplement 1A*), or 'delayed-sustained', starting after 100 ms post-stimulus and decaying slowly over the next 1–2 s (11/29) (*Figure 3—figure supplement 1C*). The remaining 3/29 neurons were non-responsive (*Figure 3—figure supplement 1D*). Regardless of the response to reward cues, all three groups showed significant excitations to shock cues. Notably, neurons with phasic activations to reward cue showed similar magnitude of excitations to reward cues, neutral cues, and shock cues (F = 0.9808, p>0.05, one-way ANOVA) (*Figure 3—figure supplement 1A*), suggesting that these neurons respond to stimuli in a non-specific manner. In contrast, neurons showing delayed-sustained activations by reward cues also showed persistent excitation during the presentation of shock cues that was not present after neutral cues, suggesting that these neurons are activated by stimuli of either positive or negative (but not neutral) motivational value, consistent with salience-like encoding.

## LHb neurons encode motivational valence

Our cFos results are consistent with an rEPN role in driving LHb responses to motivational stimuli, but electrophysiological recordings suggested a possibly more complex pattern. To more directly test the rEPN influence on the LHb, we recorded LHb neurons in a separate group of rats before and after inactivating the ipsilateral EPN via micro-infusing a cocktail of $GABA_A/GABA_B$ receptor agonists (0.05 nmol muscimol and 0.5 nmol baclofen in 0.3 ml PBS) over the course of 2 min during the latter half of each recording session (*Figures 4A,B* and *5A,B*). Reward and shocks were delivered with no omissions. We noted that ipsilateral EPN infusion did not affect food approach behavior during either reward trials and shock trials (*Figure 4C*).

Prior to rEPN inactivation, we found, as expected, that large proportions of LHb neurons were inhibited by the reward cue (20/32) between 200 and 400 ms post stimuli and were activated by shock cues (19/32) and footshocks (15/32) between 0 and 100 ms post stimuli (p<0.05, Mann-Whitney U test) (*Figure 4D*), similar to the predominant response pattern we observed in the rEPN. Furthermore, we observed monotonic coding of value in LHb responses to predictive cues during 0–500 ms post-stimulus window (p=0.0034 and p=0.0085 for reward vs. neutral cues and neutral vs. shock cues respectively, repeated measures one-way ANOVA, Holm-Sidak correction for multiple comparisons) (*Figure 4F*), which is consistent with previous studies showing that LHb neurons encode motivational valence (*Hong and Hikosaka, 2008*; *Jhou et al., 2013*; *Matsumoto and Hikosaka, 2007*). However, although responses to shock cues correlated positively with individual neuron responses to footshocks ($r^2$ = 0.2027, p=0.01), we found that only 8/32 LHb neurons were significantly excited by both shocks and cues, while 10/32 and 7/32 LHb neurons showed significant

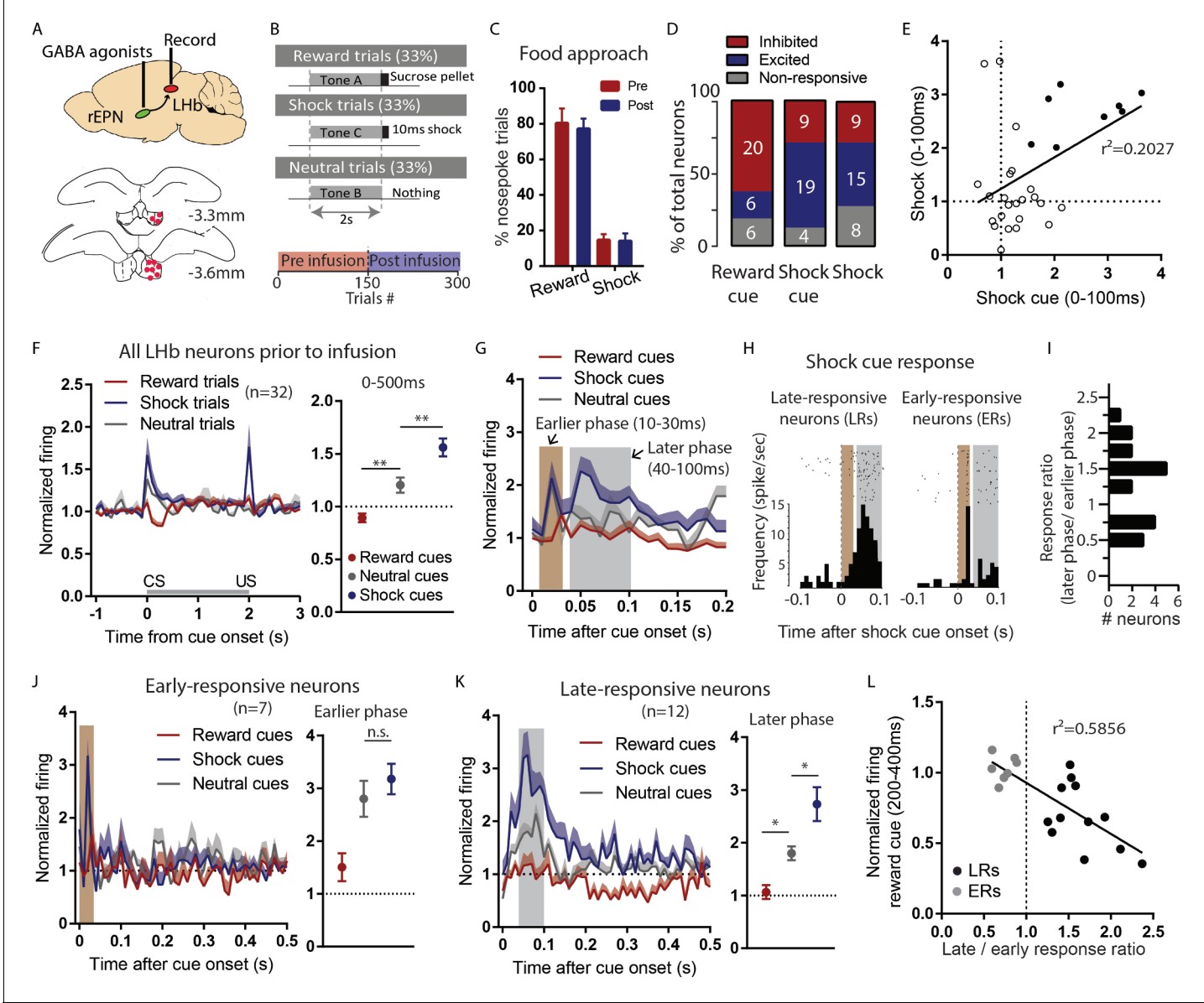

**Figure 4.** LHb responses to shock cues are encoded in two distinct subpopulations. (**A**) Schematic of recording from the LHb while inactivating ipsilateral rEPN with GABA agonists. (**B**) Schematic of inactivation behavioral paradigm in which reward and shock were delivered at 100% probability. (**C**) Ipsilateral rEPN inactivaiton did not affect behavioral response (food tray approach) in reward trials. (**D**) A majority of LHb neurons are inhibited by reward cue and activated by shock and shock cues. (**E**) Responses to shock cues were positively correlated with individual neuron's responses to footshocks. Solid circles indicate neurons significantly excited by both shock cues and footshocks. (**F**) LHb neurons on average showed inhibition to reward cues and excitation to shock cues during 0–500 ms post-stimulus window, suggesting a valence encoding pattern. (**G**) Excitation to shock cues contained two phases: an 'earlier' phase from 10 to 30 ms post-stimulus and a 'later' phase from 40 to 100 ms post-stimulus (brown and gray shaded bars, respectively). (**H**) Rasterplots and histograms of representative early- and late-responsive LHb neurons. (**I**) Histogram of all shock cue-activated LHb neurons shows bimodal distribution of ratio late to early response components. (**J**) Early-responsive neurons do not differentiate between shock cues and neutral cues. (**K**) In contrast, late-responsive neurons exhibited greater responses to shock cues compared to neutral cues and reward cues. Shaded bar indiactes analysis windows for ajacent panels. (**L**) Late shock cue-responsive showed strongly inhibitions to reward cues than early shock cue-responsive neurons.

DOI: https://doi.org/10.7554/eLife.41223.007

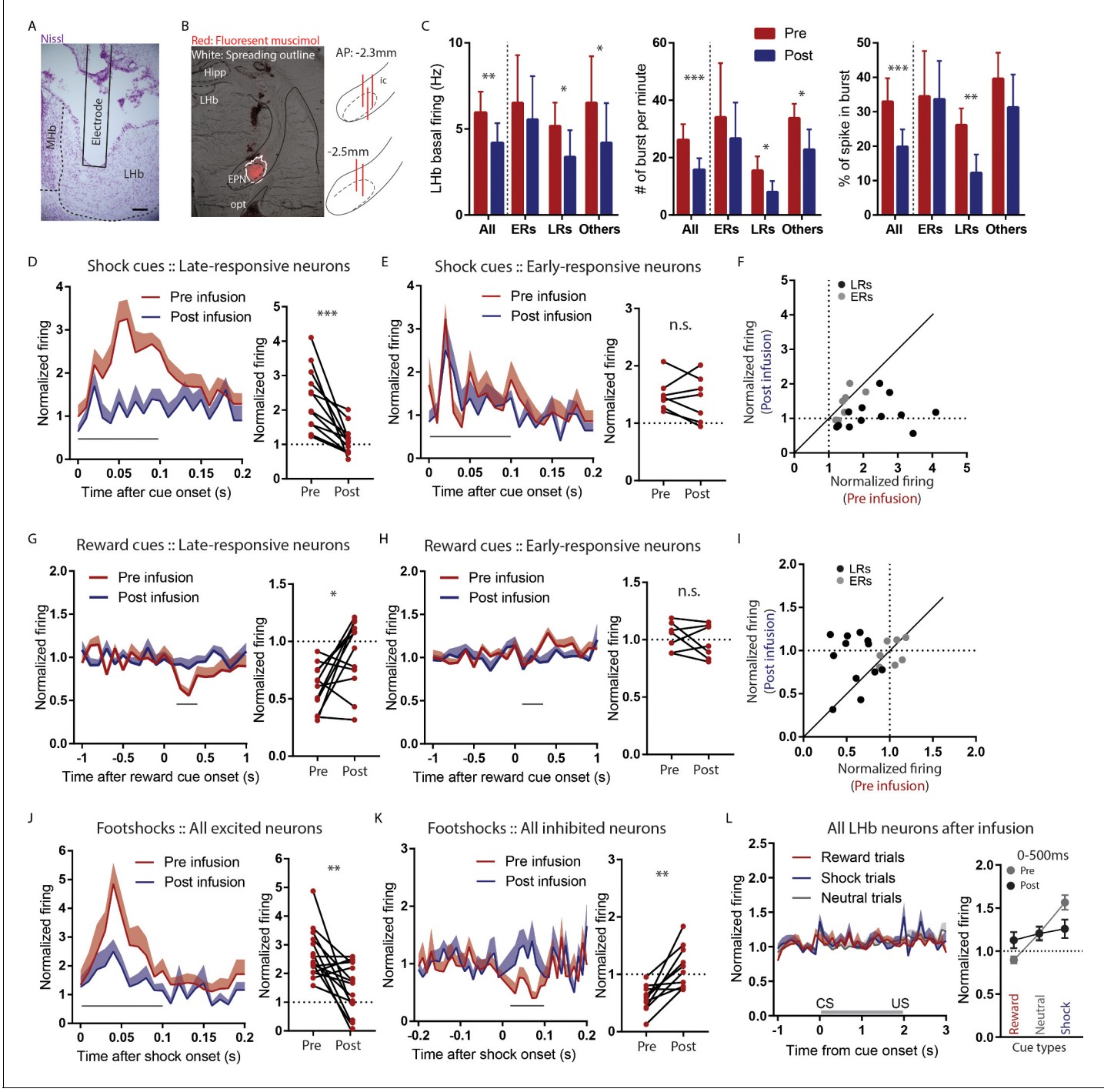

**Figure 5.** Temporary inactivation of the rEPN reduces LHb basal firings and preferentially diminishes LHb responses to negative motivational stimuli. (A) Representative photomicrograph of recording electrode track in the LHb. Scalebar: 100 μm. (B) Representative photo of muscimol spread over 30 min and cannula placements in the EPN. (C) rEPN inactivation decreased basal firing, number of bursts per minute, and percentage of spikes in bursts in LHb neurons. (D–F) Responses of late- but not early-responsive LHb neurons to shock cues were eliminated by rEPN inactivation. (G–I) Responses of late- but not early-responsive LHb neurons to reward cues were eliminated by rEPN inactivation. (J, K) rEPN inactivation greatly diminished both excitatory and inhibitory LHb responses to footshocks. Black bars under time course indicate analysis windows for adjacent panels. (L) After rEPN inactivation, the population average of LHb responses only showed weak responses to cues that did not discriminate between different cue types (compared with robust responses in *Figure 4F*). Adjacent panel on right shows loss of monotonic responses to shock cue vs neutral cue vs reward cue (black symbols), relative to pre-infusion responses (grey symbols, adapted from *Figure 4F*).

DOI: https://doi.org/10.7554/eLife.41223.008

excitations to only shock cues or footshocks (*Figure 4E*), similar to the separate encoding of CS and US we had seen in rEPN neurons.

## LHb responses to shock cues occupy two distinct temporal phases in largely distinct subpopulations

When further analyzing LHb responses to shock cues in 10 ms bins, we found that these responses consisted of two temporal phases, an 'earlier' phase occurring 10–30 ms post-stimulus and a 'later' phase occurring 40–100 ms post-stimulus (*Figure 4G*). In contrast, neutral cues only produced an earlier phase activation and reward cues caused no activation (*Figure 4G*). We further found that the two phases of response were contained in separate populations of LHb neurons. In particular, we identified 'early-responsive' neurons (7/19) that showed strong excitations during the earlier phase and relatively weaker or no responses during the later phase, while 'late-responsive' neurons (12/19) exhibited larger excitations during the later than earlier phases after shock cues (*Figure 4H*). The separation between these types of neurons was further evidenced by examining the ratio of response magnitudes of the late to early responses to shock, showing a bimodal distribution of this ratio (*Figure 4I*). These two neuron groups appear to encode very different types of information. In particular, early-responsive neurons responded with similar magnitudes to shock cues and neutral cues (F = 4.88, p=0.36, one-way ANOVA) (*Figure 4J*), while late-responsive neurons exhibited significantly greater responses to shock cues than neutral cues, and greater responses to neutral cues than reward cues during a 0–100 ms post-stimulus window (F = 2.137, p=0.0046 and p=0.04 for reward cues vs. neutral cues and neutral cues vs. shock cues, respectively) (*Figure 4K*). Additionally, 8/12 late-responsive neurons were significantly inhibited by reward cues, while none of the early-responsive neurons were inhibited, and the ratio of the late to early phase responses to shock cues correlated negatively with individual neuron's response to reward cues, suggesting that late shock cue-responsive neurons preferentially encode inhibitory response to reward cues ($r^2$ = 0.5856, p=0.0001) (*Figure 4L*).

Hence, we observed two broad categories of LHb neurons potentially encoding different information at different times, with late but not early shock cue-responsive neurons encoding valence.

## Temporary inactivation of the rEPN reduces LHb basal firings and diminishes LHb responses to motivational stimuli preferentially in late-responsive LHb neurons

After EPN inactivation, we observed dramatic decreases in basal activities of LHb neurons. Specifically, EPN inactivation significantly reduced LHb basal firing from 5.95 to 4.18 Hz, reduced bursts per minute from 26.26 to 15.8, and reduced percentages of spikes in bursts from 32.5 to 23.97 (t = 3.569, p=0.0019; t = 4.698, p<0.0001; t = 3.621, p=0.0014, respectively, paired t-test) (*Figure 5C*). Interestingly, basal activities of late-responsive LHb neurons were greatly affected by the inactivation, while early-responsive LHb neurons were mostly unaffected (pre v.s. post in basal firing, ERs: p=0.321, LRs: p=0.047, others: p=0.03; in bursts per minute, ERs: p=0.09, LRs: p=0.012, others: p=0.027; in percentages of spikes in bursts, ERs: p=0.312, LRs: p=0.009, others: p=0.14, two-way ANOVA, Holm-Sidak correction for multiple comparisons) (*Figure 5C*).

Since late-responsive neurons were more sensitive to EPN inactivation than early-responsive neurons and preferentially encoded valence, we further examined effects of EPN inactivation on valence encoding. We found that responses to shock cues in late-responsive neurons were almost eliminated by rEPN inactivation (interaction: p<0.0001, repeated measure two-way ANOVA; p=0.00080, paired t-test), while early-responsive neurons remained unaffected (interaction: p=0.634, repeated measure two-way ANOVA; p=0.538, paired t-test) (*Figure 5D,E*), and the inactivation-induced changes in individual neuron responses to shock cues correlated negatively with their ratio of the late to early phase responses ($r^2$ = 0.5028 and p=0.0007) (*Figure 5F*). Furthermore, EPN inactivation also abolished LHb inhibitory responses to reward cues in late-responsive neurons (interaction: p=0.052, repeated measure two-way ANOVA; p=0.016, paired t-test), but not early-responsive neurons (interaction: p=0.291, repeated measure two-way ANOVA; p=0.4013, paired t-test) (*Figure 5G,H*). Again, inactivation-induced changes in individual neuron responses to reward cues positively correlated with their ratio of the late to early phase responses ($r^2$ = 0.3861 and p=0.0045) (*Figure 5I*).

We further observed that LHb activation to footshocks themselves did not exhibit two clear temporally separated response phases as had been seen for shock cues, and found that rEPN inactivation significantly reduced magnitude in 10 out of 15 shock-activated neurons (interaction: p=0.391, repeated measure two-way ANOVA; p=0.0032, paired t-test) (*Figure 5J*). In particular, we observed the reductions in five out of eight late-responsive neurons, as well as in seven neurons that did not respond to shock cues, but not in early-responsive neurons. Additionally, we and previous studies have noted a small proportion of LHb neurons that are inhibited by footshocks (*Lecca et al., 2017*; *Wang et al., 2017*), and we further found that these responses were also significantly reduced by rEPN inactivation (interaction: p=0.1254, repeated measure two-way ANOVA; p=0.002, paired t-test) (*Figure 5K*). Furthermore, we observed that rEPN inactivation completely abolished valence encoding in LHb neurons, as population LHb responses no longer discriminated between reward, neutral, and shock cues (*Figure 5L*) (F = 1.21, repeated measures one-way ANOVA, p=0.145 and p=0.709 for reward vs. neutral cues and neutral vs. shock cues respectively, Holm-Sidak correction for multiple comparisons).

## Excitotoxic lesions of the rEPN reduce footshock-induced cFos in LHb and RMTg

We next tested whether the rEPN was necessary to drive footshock-induced immediate early gene expression in the RMTg. We unilaterally lesioned the rEPN with an excitotoxin and measured the ratio of ipsilateral over contralateral cFos expression in the LHb and RMTg in response to a series of

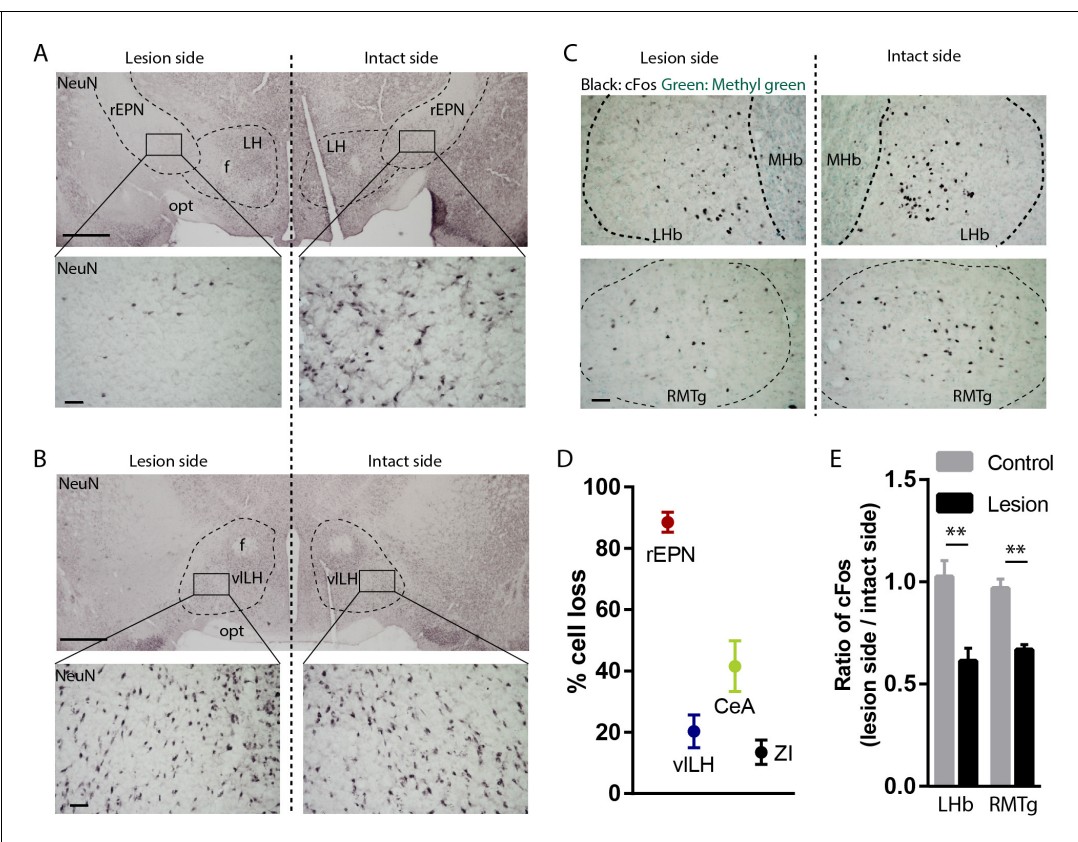

**Figure 6.** Excitotoxic lesion of the rEPN reduces cFos induced by unsignaled footshocks in LHb and RMTg. (**A, B**) Representative photomicrographs of the ipsilateral (lesioned) and contralateral (intact) rEPN and vlLH immunostained for the neuronal marker NeuN (black label). (**C**) Representative photographs of footshock-induced cFos (black label) in the LHb and RMTg with methyl green counterstain. (**D**) Number of cells dramatically decreased in the entire rEPN, on the lesioned side, with smaller reductions in vlHL, CeA and ZI. (**E**) rEPN lesion reduced cFos expression induced by unsignaled footshocks in the ipsilateral LHb and RMTg, compared with contralateral (intact) side. Scalebars are 1 mm in top panels for **A**), (**B**), and 100 µm in all other panels.
DOI: https://doi.org/10.7554/eLife.41223.009

footshocks (n = 4 for lesion group and n = 4 for control group). NeuN staining (*Figure 6A,B*) showed a greater than 85% loss of rEPN cells in the lesioned side compared with the intact side, along with smaller cell losses in the adjacent vlLH, rostral portion of the central nucleus of amygdala (CeA) and zona incerta (ZI) (p<0.0001, one-way ANOVA) (*Figure 6D*). rEPN lesions also significantly diminished footshock-induced cFos in the ipsilateral LHb and RMTg by 42% and 38%, respectively (F = 13.68, one-way ANOVA, p=0.0014, p=0.008 for the LHb and the RMTg respectively between lesion and intact). In contrast, sham-lesioned animals showed equal amounts of cFos on both sides (*Figure 6C, E*). These data further suggest that the rEPN plays a partial role in driving LHb and RMTg responses to footshocks.

## Discussion

In the present study, we identified the rEPN as a particularly selective source of anatomic afferents to RMTg-projecting (versus VTA-projecting) LHb neurons. We then found that LHb neurons exhibit heterogeneous responses to motivational stimuli, and that the rEPN contributes to LHb basal firing and responses to motivational stimuli particularly in a subset of 'late-responsive' LHb neurons that have slower phasic responses to aversive stimuli and exhibit valence-encoding firing patterns. rEPN inactivation also reduced burst firing in these neurons, while rEPN lesions reduced but did not eliminate footshock-induced cFos in the LHb and RMTg.

Previous studies have indicated that genetically distinct rEPN neurons project to different targets and exhibit distinct electrophysiological patterns, with somatostatin positive neurons projecting to the LHb and showing inhibitory responses to reward cues (*Stephenson-Jones et al., 2016*; *Wallace et al., 2017*). Consistent with these and other studies (*Hong and Hikosaka, 2008*), we showed that reward cue-inhibited rEPN neurons encoded motivational valence (shock cue >neutral cue>reward cue) and that these neurons were most likely to also be activated by footshocks and reward omissions. In contrast, we found that reward cue-excited or non-responsive rEPN neurons did not encode motivational valence, and instead showed a variety of other encoding patterns more consistent with encoding of salience, or of a non-specific response to all auditory cues.

Similar to reward cue-inhibited neurons in the rEPN, a majority of LHb neurons also exhibited inhibitory response to reward cues and excitatory responses to shock and shock cues, which is generally consistent with a valence-encoding pattern reported previously (*Hong et al., 2011*; *Matsumoto and Hikosaka, 2007*). However, we further showed that these valence-encoding properties (inhibition to reward cues, scaled responses between neutral and shock cues, and excitation to footshocks) were concentrated in a specific subpopulation of LHb neurons (late shock cue-responsive LHb neurons). Although we did not test LHb neurons for encoding of prediction errors (e.g. reward omission), we speculate that the late-responsive neurons would also show prediction error signals based on their responsiveness to other valence signals, and also based on their similarity to rEPN valence-encoding neurons which were activated by reward omission. Interestingly, it is notable that CS and US activations were largely encoded in separated populations in both rEPN and LHb with only roughly one third of CS or US activated neurons in both structures encoding both stimuli, suggesting that CS and US responses in rEPN and LHb neurons are mediated by parallel inputs. Other results from our lab suggest that these signals remain separate in the RMTg (Li, Vento and Jhou, unpublished findings), and do not converge until they reach downstream targets such as the VTA (*Cohen et al., 2012*).

Most critically, we demonstrated that the bulk of LHb encoding of valence was dependent on the rEPN, as rEPN inactivation selectively eliminated responses to both positive and negative motivational stimuli (cues and outcomes) in either late-responsive LHb neurons or in all recorded LHb neurons. These results are consistent with a recent study showing a bidirectional modulation of the rEPN-LHb pathway on reinforcement learning (*Stephenson-Jones et al., 2016*). In that study, the authors used a task in which animals performing nosepokes at two ports received either a reward or non-reward (omission). They found that either activating or inhibiting the rEPN-LHb pathway or either reducing GABA or glutamate receptors in rEPN neurons are sufficient to discourage or encourage reinforcement learning in a decision-making task, suggesting that the rEPN-LHb pathway mediates both positive and negative motivations. Our results provide further evidence that rEPN modulation by either positive and negative motivational stimuli could influence the LHb. Notably, the effects of rEPN inactivation, especially the fact that some inhibitory LHb responses were also

eliminated by rEPN inactivation, are complicated by the presence of GABA/glutamate co-releasing neurons in the rEPN (*Meye et al., 2016*; *Root et al., 2018*; *Wallace et al., 2017*). Previous studies have pointed out that rEPN neurons can either excite or inhibit LHb neurons depending on the balance and synaptic strength between glutamatergic and GABAergic transmissions (*Li et al., 2011*; *Meye et al., 2016*). Hence, it is possible that some rEPN neurons that preferentially release GABA could in turn inhibit LHb neurons. However, the contributions of the different neurotransmitters released by rEPN neurons are still to be determined.

Interestingly, we noticed that in contrast to LHb discriminatory responses to cues that are largely absent after rEPN inactivation, LHb responses to footshocks were still present after rEPN inactivation. Similarly, rEPN lesions only partly decreased footshock-induced cFos in the RMTg. This suggests that afferents to the LHb arising from outside the lesioned or muscimol-infused region might be driving these residual responses to shocks. In the lesion experiment, about 68% of the response was still present, which could have been driven by the vlLH, which contained a subset of footshock-activated LHb-projecting neurons that were largely spared by the lesion. For the muscimol experiments, only 35% of the shock response remained, with this more complete reduction possibly being due to the muscimol infusion being relatively large (300 nl) and hence spreading somewhat further to the immediately adjacent hypothalamus. Although florescent muscimol infused into the rEPN remained confined in that region, the recording sessions used non-fluorescent muscimol having a five-fold lower molecular weight (114.1 v.s. 607.5) that likely would have diffused farther. Hence, we cannot exclude the possibility that regions other than the rEPN, such as vlLH, also play roles in mediating valence-encoding in the LHb. Consistent with this, recent studies have found aversive effects of activating numerous input to the LHb arising from the LH, VTA, VP, and LPO (*Barker et al., 2017*; *Lecca et al., 2017*; *Root et al., 2014*; *Stamatakis et al., 2016*; *Tooley et al., 2018*). Although we did not observe shock-induced cFos in most of these regions, the presence of mixed GABA and glutamate cells projecting to the LHb from each of these regions could potentially have obscured an effect.

In addition to the rEPN contribution to LHb responses to affective stimuli, the present study also found a large rEPN contribution to LHb basal firing and bursting patterns. Numerous studies have indicated the close relationship between elevated LHb basal and bursting firings and depressive symptoms (*Cui et al., 2018*; *Lawson et al., 2017*; *Shumake and Gonzalez-Lima, 2003*; *Yang et al., 2018*). Here, we showed that rEPN inactivation dramatically decreased LHb basal firing rates and reduced bursting firing patterns, suggesting that the rEPN also contributes to burst firing patterns in the LHb, and potentially plays a critical role of the rEPN in regulating depressive symptoms (*Baker et al., 2016*; *Li et al., 2011*; *Shabel et al., 2014*).

In conclusion, we demonstrate the important role of rEPN neurons, and possible adjacent regions, in mediating LHb basal firings and responses to motivational stimuli. Our study potentially provides novel insights into processing of motivational stimuli, which may be disrupted in many psychiatric disorders.

## Materials and methods

### Animals

All procedures were conducted under the National Institutes of Health Guide for the Care and Use of Laboratory Animals, and all protocols were approved by Medical University of South Carolina Institutional Animal Care and Use Committee. Adult male Sprague Dawley rats weighing 250 to 450 g from Charles River Laboratories were paired housed in standard shoebox cages with food and water provided ad libitum until experiments started. Rats were single housed during all experiments. In total, 38 rats were used for these experiments. Sixteen rats completed recordings. Of these, 10 rats were used for rEPN recoding, and six rats for LHb recordings. Fourteen underwent CTb cFos experiments with seven rats for footshock group and seven rats for control group. eight rats underwent rEPN lesion cFos experiments with four rats for lesion group and four rats for control group.

### Surgeries

All surgeries were conducted under aseptic conditions with rats that were under isoflurane (1–2% at 0.5–1.0 l/min) anesthesia. Analgesic (ketoprofen, 5 mg/kg) was administered subcutaneously

immediately after surgery. Rats were given at least 5 days to recover from surgery. For rabies tracing experiments, 300 nl of retroAAV-cre (Addgene) was injected ipsilaterally with a glass pipette into the VTA (AP: −5.2 mm; ML: 2 mm; DV: −8.2 mm from dura, 10-degree angle) or the RMTg (AP: −7.4 mm; ML: 2.1 mm; DV: −7.7 mm from dura, 10-degree angle). Flexed-TVA-mcherry and Flexed-RG into the LHb (AP: −3.4 mm; ML: 1.5 mm; DV: −4.5 mm from dura, 10-degree angle) at the same time. 21 days later, EnvA-ΔG-rabies-GFP was injected into the LHb. For recording experiments, custom drivable electrode arrays were implanted above the rEPN (AP: −2.3 mm; ML: 2.8 mm; DV: −7.1 mm from dura) or the LHb (AP: −3.4 mm; ML: 1.5 mm; DV: −4.5 mm from dura, 10-degree angle). For CTb injection, 80 nl of CTb was injected ipsilaterally into the LHb. For lesion experiments, 50 nl of 400 mM quinolinic acid per side was injected into the rEPN (AP: −2.5 mm; ML: 3.0 mm; DV: −7.5 mm from dura). Rats were kept anesthetized with pentobarbital intraperitoneally (55 mg/kg) for up to 3 hr' post-surgery to reduce excitotoxic effect.

## Perfusions and tissue sectioning

Rats used for all experiments were sacrificed with an overdose of isoflurane and perfused transcardially with 10% formalin in 0.1M phosophate buffered saline (PBS), pH 7.4. Brains from electrophysiology experiments had passage of 100 μA current before perfusion, allowing electrode tips to be visualized. Brains were removed from the skull, and post-fix in 10% formalin for 24 hr before equilibrated in 20% sucrose solution until sunk. Brains were cut into 40 μm sections on a freezing microtome. Sections were stored in phosphate buffered saline with 0.05% sodium azide.

## Immunohistochemistry

Free-floating sections were immunostained for CTb, NeuN or cFos by overnight incubation in goat anti-CTb (List Biological Laboratories, 7032A9, 1: 50,000 dilution), mouse anti-NeuN (Millipore, MAB-377, 1: 5000 dilution), or rabbit anti-cFos (Millipore, ABE457, 1:1000 dilution) primary in PBS with 0.25% Triton-X and 0.05% sodium azide. Afterwards, tissue was washed three times in PBS and incubated in biotinylated donkey-anti-goat, anti-mouse or anti-rabbit secondary (1:1000 dilution, Jackson Immunoresearch, West Grove, PA) for 30 min, followed by three 30 s rinses in PBS, followed by 1 hr in avidin-biotin complex (Vector). For TH-staining, tissue was then rinsed in sodium acetate buffer (0.1M, pH 7.4), followed by incubation for 5 min in 1% diaminobenzidine (DAB). For cFos and NeuN staining, nickel and hydrogen peroxide (Vector) were added to reveal a blue-black reaction product.

## Behavioral training for electrophysiological recordings

Rats were food restricted to 85% of their *ad libitum* body weight and trained to associate distinct auditory cues with either a sucrose pellet or no outcome. Behavior was conducted in standard Med Associates chambers (St. Albans, VT). Reward and neutral cues were a 1 kHz tone (75 dB) and white noise (75 dB), respectively. The Reward cue was presented for 2 s, and a sucrose pellet (45 mg, Bio-Serv) was delivered immediately after cue offset. The neutral cue was also presented for 2 s, but no sucrose pellet was delivered. The two trial types were randomly presented with a 30 s interval between successive trials. A 'correct' response was scored if the animal either entered the food tray within 2 s after reward cues, or withheld a response for 2 s after neutral cues. Rats were trained with 100 trials per session, one session per day, until they achieved 85% accuracy in any 20-trial block. Once 85% accuracy was established, rats underwent surgeries. After recovery from surgeries, rats were then trained with one extra session in which neutral cue trials were replaced by aversive trials consisting of a 2 s 8 kHz tone (75 dB) followed by a 10 ms 0.6mA footshock.

For Pavlovian conditioning paradigm, once rats achieved 85% accuracy in reward trials, they were trained to respond to an 8 kHz tone (75 dB) lasting for 2 s followed by a mild footshock (0.6mA). During testing, rats again placed on mild food deprivation, and recordings obtained in sessions consisting of 150 mixture of reward trials, neutral trials, and shock trials randomly selected. Rats were recorded for one or two session per day, and electrodes advanced 80–160 μm at the end of each session. Neurons with significant reductions in baseline firing rates across sessions were excluded from the study, as this is indicative of drifting of recording wires between sessions.

## Electrophysiological recordings

After final training, electrodes consisted of a bundle of sixteen 18 µm Formvar-insulated nichrome wires (A-M system) attached to a custom-machined circuit board. Electrodes were grounded through a 37-gauge wire attached to a gold-plated pin (Newark Electronics), which was implanted into the overlying cortex. Recordings were performed during once-daily sessions, and electrodes were advanced 80–160 µm at the end of each session. The recording apparatus consisted of a unity gain headstage (Neurosys LLC) whose output was fed to preamplifiers with high-pass and low-pass filter cutoffs of 300 Hz and 6 kHz, respectively. Analog signals were converted to 18-bit values at a frequency of 15.625 kHz using a PCI card (National Instruments) controlled by customized acquisition software (Neurosys LLC). Spikes were initially detected via thresholding to remove signals less than twofold above background noise levels, and signals were further processed using principal component analysis performed by NeuroSorter software. Spikes were accepted only if they had a refractory period, determined by <0.2% of spikes occurring within 1 ms of a previous spike, as well as by the presence of a large central notch in the auto-correlogram. Neurons that had significant drifts in firing rates and that showed 10% or above similarity in cross-correlogram were excluded. Since the shock duration used in the present study was 10 ms, the first 10 ms of data after footshock were removed in order to reduce shock artifacts.

## cFos shock induction

All animals were habituated in the behavioral chamber 20 min for 3 days. In the last day, animals received 30 footshocks lasting 5–15 s (0.7mA). The induction process lasted 20 min. Animals were perfused 1 hr after the end of cFos induction program.

## Statistical analysis of electrophysiological and behavioral data

To increase statistical power, recording data includes both technical replicates (multiple trials during which individual neurons were recorded) and biological replicates (multiple subjects).

Burst analysis was performed with NeuroExplorer, with a burst defined by any clusters of spikes beginning with a maximal inter-spike interval of 20 ms and ending with a maximal inter-spike interval of 100 ms. The minimum intra-burst interval was set at 100 ms and the minimum number of spikes in a burst was set at 2. Burst per minute, the percentage of spike firing in bursts, and mean duration of bursts were analyzed.

Neurons with large drifting of the microwire electrodes during recordings were excluded from further analysis. Electrophysiology data were first tested for normality, then transformed to ranked forms if data failed tests of normality ($p < 0.05$, D'Agostino-Pearson test). Significant responses in neural firing were determined by a threshold of $p < 0.05$ for each neuron's firing rate versus baseline (Mann-Whitney U test). Data were tested for normality ($p > 0.05$, D'Agostino-Pearson test) and were analyzed using parametric tests. One-way or two-way ANOVA with Holm-Sidak correction for multiple comparisons, and two-tailed t-test were used to compare across experimental conditions, respectively, if not otherwise specified. Calculations were performed using Matlab (Mathworks) and Prism seven software (Graph Pad).

## Additional information

### Funding

| Funder | Grant reference number | Author |
| --- | --- | --- |
| National Institutes of Health | DA037327 | Thomas C Jhou |
| National Institutes of Health | DA032898 | Thomas C Jhou |

The funders had no role in study design, data collection and interpretation, or the decision to submit the work for publication.

### Author contributions

Hao Li, Conceptualization, Data curation, Formal analysis, Investigation, Methodology, Writing—original draft, Writing—review and editing; Dominika Pullmann, Data curation, Methodology, Project

administration; Thomas C Jhou, Conceptualization, Supervision, Funding acquisition, Writing—review and editing

### Author ORCIDs

Thomas C Jhou (iD) http://orcid.org/0000-0001-8811-0156

### Ethics

Animal experimentation: This study was performed in strict accordance with the recommendations in the Guide for the Care and Use of Laboratory Animals of the National Institutes of Health. All of the animals were handled according to approved institutional animal care and use committee (IACUC) protocols (2988) of the Medical University of South Carolina. The protocol was approved by the Institutional Animal Care and Use Committee of the Medical University of South Carolina (DHHS Assurance #A3428-01). All surgery was performed under isoflurane anesthesia, and every effort was made to minimize suffering.

### Decision letter and Author response

Decision letter https://doi.org/10.7554/eLife.41223.012
Author response https://doi.org/10.7554/eLife.41223.013

## Additional files

### Supplementary files

• Transparent reporting form
DOI: https://doi.org/10.7554/eLife.41223.010

### Data availability

All data generated or analyzed during this study are included in the manuscript and supporting files.

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
