## [Decision Letter]

Thank you for submitting your article "The entopeduncular nucleus drives lateral habenula responses to negative but not positive or neutral affective stimuli" for consideration by *eLife*. Your article has been reviewed by three peer reviewers, and the evaluation has been overseen by a Reviewing Editor and Michael Frank as the Senior Editor. The following individuals involved in review of your submission have agreed to reveal their identity: Masayuki Matsumoto (Reviewer #2); Mitsuko Watabe-Uchida (Reviewer #3).

The reviewers have discussed the reviews with one another and the Reviewing Editor has drafted this decision to help you prepare a revised submission.

Summary:

The lateral habenula (LHb) has been thought to be a key brain structure that regulates reinforcement learning and mood. LHb neurons signal negative reward prediction errors but how their activity is generated remains to be clarified. A previous study identified neurons in the entopeduncular nucleus (EPN, the rodent homolog of globus pallidus internal segment) as a candidate input that regulates 'phasic' responses of LHb neurons (Stepheson-Jones et al., 2016). In the present study, Li and colleagues tested this idea by manipulating the activity of EPN neurons while recording single unit activities in the LHb.

Temporary inactivation of the rostral EPN (rEPN) neurons using muscimol caused a decrease in the baseline activity in LHb neurons. Furthermore, LHb neurons' responses to shock-predicting cues and shock were greatly reduced whereas the inhibitory response to reward predicting cues was largely spared. The authors conclude that rEPN plays an important but selective role in the generation of LHb responses to motivational stimuli.

All the reviewers thought that the authors address a timely and important question, performed heroic experiments, and provide interesting results. Although the reviewers are positive overall, they also raised a number of substantive concerns that need to be addressed before publication.

Essential points:

1) Selective role. In their analysis the authors show that the normalized inhibitory responses to reward in the LHb are not significantly different between pre and post infusions. However, given that inhibiting the rEPN reduces the LHb baseline by half the absolute reduction in firing rate by definition also has to be half in order for the normalized response amplitudes to be the same. Given this then one could make the claim that 50% of the inhibitory LHb response to the reward predicting cues comes from the rEPN. The authors only use the normalized firing rates to make a claim that there is no difference in the inhibitory response but if they used absolute firing then the opposite claim that the rEPN contributes significantly to the LHb inhibitory response could be made. Due to the large change in baseline firing rate and the fact an argument could be made for or against an influence of on the inhibitory response there is no conclusive way that the authors can make a large claim about the contribution of the rEPN to inhibitory responses in the LHb from their data. We do appreciate that the data clearly shows that the rEPN is not the sole source of the inhibitory response to reward predicting cues but we do not think the authors have shown that it doesn't have an effect.

The only way to address this problem is by preventing the rEPN neurons from being inhibited without influencing their baseline firing rate. This was attempted by Stephenson-Jones et al., 2016, by knocking down the GABA_A_ receptors in rEPN neurons. When this was done the effect was a significant reduction in the ability of mice to process positive but not negative (i.e. omissions) feedback. This suggests that the inhibition of the rEPN does have a significant contribution to behavior. Indeed, in the same paper authors showed that inhibiting the rEPN is rewarding and can reinforce behavior. Suggesting that it does have an effect on the LHb.

Overall, the main conclusion (a specific effect of EP inactivation on LHb responses to shock cue) is not convincing because the present analysis solely relies on a comparison using a single significance test. From the figure shown in Figure 4D and G, it looks like LHb neurons lost not only the excitation by shock cues but also the inhibition by reward cues (is the deviation from baseline 1.0 significant)? It is also useful to show how many neurons out of 18 lost excitation and how many out of 20 lost inhibition and how is value coding (reward cue < neutral cue < shock cue) as a population (not just average) affected by inactivation. Because authors have single unit data, conclusion can be examined with multiple methods.

The interpretation of the rEPN's contribution to inhibitory LHb responses could be complicated. However, we encourage the authors to provide more analyses to fully describe the results. The authors should more carefully describe the results of inactivation beyond the normalized firing rates, and the results need to be discussed more carefully. We suspect that it is difficult to firmly conclude a 'selective' role given the large baseline change. If this is the case, we suggest that the authors reduce the tone of a 'selective' role throughout the manuscript, and restate the main conclusion.

2) Inactivation was done pharmacologically, but there is no precise description of the method in the manuscript. Because there are important nuclei surrounding EP, such as zona incerta, lateral hypothalamus, subthalamic nucleus and amygdala, it is important to estimate the spread of the solution during the session. For example, if the session lasts for 30 min, please estimate the spread of the drug using injection of fluorescence muscimol and fixation after 30 min (although this would underestimate spread because fluorescence muscimol is much bigger than muscimol). If there is some leak, control experiments with drug injection at the nearby location will be necessary.

3) Behavioral performance to confirm learning. The authors do not show any behavioral data, such as licking or blinking, during the conditioning. Such behavioral data are necessary to prove that the animals understood the association between cue and outcome. In addition, licking or blinking might change after rEPN inactivation. Such behavior data would be helpful to consider the role of rEPN in behavior control.

[Editors' note: further revisions were requested prior to acceptance, as described below.]

Thank you for resubmitting your article "The entopeduncular nucleus drives valence-encoding in the lateral habenula" for consideration by *eLife*. Your revised article has been reviewed by three peer reviewers, and the evaluation has been overseen by a Reviewing Editor and Michael Frank as the Senior Editor. The following individuals involved in review of your submission have agreed to reveal their identity: Marcus Stephenson-Jones (Reviewer #1); Masayuki Matsumoto (Reviewer #2); Mitsuko Watabe-Uchida (Reviewer #3).

The reviewers have discussed the reviews with one another and the Reviewing Editor has drafted this decision to help you prepare a revised submission.

The reviewers agreed that the authors have addressed most of their previous concerns. However, some issues remain to be addressed. We think that these concerns can be addressed without new experiments. We would like the authors to address the remaining concerns before publication of this study at *eLife*.

*Reviewer #1:*

The authors have addressed my major concerns and the manuscript is significantly improved because the data now support their major conclusions. I also appreciate the fact the authors now briefly discuss additional potential sources of valence information to the LHb from the VP, VTA etc. In general, they have addressed my concerns and I just have a few minor issues.

1) The authors note that 50nl of quinolinic acid reduced the number of cFos positive cells after footshocks in the dlLH as well as the rEPN. Do they not also expect the muscimol that is injected in the rEPN to also partially effect the dlLH? Related to this the authors do not mention the volume of muscimol that is injected. If it is larger than 50nl then potentially even more of the dlLH will be affected. I think it should be mentioned in the Discussion that part of the effects that are observed may have been due to adjacent areas.

2) In the rabies figure the authors superimpose images from two different mice. This gives the impression that the images come from a single mouse. It is not clear to me how two images from separate mice can accurately be overlayed. I would suggest showing the data as separate images.

3) As the authors now split their analysis between early and late responsive cells all the analysis should be done on these two groups. Some of the analysis is still done on the single population, for example the effect of the rEPN inactivation on the baseline firing is still just for the single population. I would suggest splitting the analysis.

4) Cells are reported in the rEPN and LHb that respond to either shocks or cues predicting shocks. Do these ratios change during learning? It would be nice to report if there is a difference early or late in learning as the cue response may increase and the US response decrease over the course of learning as has been reported in primates and mice.

*Reviewer #2:*

The authors properly responded to most of my concerns. I have a few further comments.

Although the authors showed behavior during reward and neutral trials (Figure 3B), I also would like to see behavior during shock trials.

Although the authors added Figure 4C showing the effect of rEPN inactivation on food approach behavior, they did not describe anything about this result. They should explain this result in the manuscript, and should discuss why the inactivation did not influence behavior.

*Reviewer #3:*

This revised manuscript by Li et al. examined effects of the mouse rostral EP (monkey GPb) on lHb. The original version did not analyze the neuronal phenotypes thoroughly (they heavily depend on single significance tests) and reviewers asked to the authors to examine this in multiple ways, including use of raw firing rates in addition to normalized firing rates, examination of population coding in addition to average, use of different time windows, and examination of RPE or value coding, etc., and to discuss carefully about those factors. In the revision, the authors found that if they divide neurons into two groups, they are able to see some effects on inhibition responses to reward, in addition to excitation responses to shock, in lHb, which was not detected in the original version. It’s good that the authors noticed that you may or may not see effects with different analyses, but I wanted to see more careful and through discussion and analyses. Just comparing normalized firing rates between pre and post muscimol in different trial types and in arbitrarily divided neurons is misleading. It is not clear how valence coding in lHb changed (or didn’t change, but just scaled down). In Figure 5F and I, here the authors also plotted ratio of post and pre-muscimol firing rates after outcomes, without considering the meaning of coding, and so lost information about inhibition or excitation, compared to baseline or neutral conditions. Overall, I think that the authors answered many of our minor comments, but really did not address our biggest concerns and increased my worries. Also, the manuscript and figures should be more focused and organized.

---

## [Author Response]

Essential points:1) Selective role. In their analysis the authors show that the normalized inhibitory responses to reward in the LHb are not significantly different between pre and post infusions. However, given that inhibiting the rEPN reduces the LHb baseline by half the absolute reduction in firing rate by definition also has to be half in order for the normalized response amplitudes to be the same. Given this then one could make the claim that 50% of the inhibitory LHb response to the reward predicting cues comes from the rEPN. The authors only use the normalized firing rates to make a claim that there is no difference in the inhibitory response but if they used absolute firing then the opposite claim that the rEPN contributes significantly to the LHb inhibitory response could be made. Due to the large change in baseline firing rate and the fact an argument could be made for or against an influence of on the inhibitory response there is no conclusive way that the authors can make a large claim about the contribution of the rEPN to inhibitory responses in the LHb from their data. We do appreciate that the data clearly shows that the rEPN is not the sole source of the inhibitory response to reward predicting cues but we do not think the authors have shown that it doesn't have an effect.The only way to address this problem is by preventing the rEPN neurons from being inhibited without influencing their baseline firing rate. This was attempted by Stephenson-Jones et al., 2016, by knocking down the GABA_A_ receptors in rEPN neurons. When this was done the effect was a significant reduction in the ability of mice to process positive but not negative (i.e. omissions) feedback. This suggests that the inhibition of the rEPN does have a significant contribution to behavior. Indeed, in the same paper authors showed that inhibiting the rEPN is rewarding and can reinforce behavior. Suggesting that it does have an effect on the LHb.Overall, the main conclusion (a specific effect of EP inactivation on LHb responses to shock cue) is not convincing because the present analysis solely relies on a comparison using a single significance test. From the figure shown in Figure 4D and G, it looks like LHb neurons lost not only the excitation by shock cues but also the inhibition by reward cues (is the deviation from baseline 1.0 significant)? It is also useful to show how many neurons out of 18 lost excitation and how many out of 20 lost inhibition and how is value coding (reward cue < neutral cue < shock cue) as a population (not just average) affected by inactivation. Because authors have single unit data, conclusion can be examined with multiple methods.The interpretation of the rEPN's contribution to inhibitory LHb responses could be complicated. However, we encourage the authors to provide more analyses to fully describe the results. The authors should more carefully describe the results of inactivation beyond the normalized firing rates, and the results need to be discussed more carefully. We suspect that it is difficult to firmly conclude a 'selective' role given the large baseline change. If this is the case, we suggest that the authors reduce the tone of a 'selective' role throughout the manuscript, and restate the main conclusion.

We appreciate that reviewer raise concerns about the selective role of the rEPN in mediating LHb responses to negative but not positive stimuli. After carefully re-analyzing our data, we agree that LHb inhibitory responses to reward cues were also affected by rEPN inactivation. However, we also noted that this effect was confined to a specific subpopulation of LHb neurons. In particular, we found that there are two temporal phases of LHb responses to shock cues, and that they occur in separate LHb neurons. “Early-responsive” neurons are activated 10-30ms post-stimulus, while “late-responsive” neurons are activated 40-100ms post-stimulus. Furthermore, late- but not early-responsive neurons contain valence-encoding characteristics, i.e. inhibitions to reward cue, excitations to shock cues, and intermediate responses to neutral cues.

After rEPN inactivation, we found that late-responsive neurons lost their valence-encoding characteristics, i.e. lost both their inhibition to reward cues and excitation to shock cues. Thus, we conclude that the rEPN has somewhat equal influence on both positive and negative motivational processing in a defined LHb subpopulation, rather than preferentially driving negative motivational processing.

We have changed title, Abstract, Results, and Discussion accordingly.

2) Inactivation was done pharmacologically, but there is no precise description of the method in the manuscript. Because there are important nuclei surrounding EP, such as zona incerta, lateral hypothalamus, subthalamic nucleus and amygdala, it is important to estimate the spread of the solution during the session. For example, if the session lasts for 30 min, please estimate the spread of the drug using injection of fluorescence muscimol and fixation after 30 min (although this would underestimate spread because fluorescence muscimol is much bigger than muscimol). If there is some leak, control experiments with drug injection at the nearby location will be necessary.

We appreciate the suggestion, and have added Figure 5B to show that the spread of fluorescent muscimol is restricted to the rEPN area when infused with the same procedure used in behavioral inactivation experiments, with animals sacrificed 30 min after the infusion (the same duration as the behavioral experiment). We agree that this might underestimate the drug spread, and have also added a note to the main text notifying the reader of this caveat.

3) Behavioral performance to confirm learning. The authors do not show any behavioral data, such as licking or blinking, during the conditioning. Such behavioral data are necessary to prove that the animals understood the association between cue and outcome. In addition, licking or blinking might change after rEPN inactivation. Such behavior data would be helpful to consider the role of rEPN in behavior control.

We agree with this concern, and have added Figure 3B showing behavioral data that confirms the cue learning. It shows that within 3 seconds of cue onset, animals approached the food port in 82% of reward trials but only 5% of neutral trials, demonstrating accurate discrimination. Importantly, the food pellet is not delivered until the cue end, so any behavior observed during cue presentation is due to the learned significance of the cues. We did not see any difference in this behavior (e.g. food port entry percentage) after rEPN inactivation (Figure 4C).

[Editors' note: further revisions were requested prior to acceptance, as described below.]

Reviewer #1:1) The authors note that 50nl of quinolinic acid reduced the number of cFos positive cells after footshocks in the dlLH as well as the rEPN. Do they not also expect the muscimol that is injected in the rEPN to also partially effect the dlLH? Related to this the authors do not mention the volume of muscimol that is injected. If it is larger than 50nl then potentially even more of the dlLH will be affected. I think it should be mentioned in the Discussion that part of the effects that are observed may have been due to adjacent areas.

We appreciate the reviewer’s concerns about off-target effects of muscimol infusion during recordings. Muscimol was infused at 300nl volume through a 26 gauge (I.D. x O.D. = 0.24mm x 0.46mm) internal cannula over the course of 2 minutes in the middle of recording sessions, as mentioned in the original draft.

We agree that muscimol infusions might have spread to adjacent areas and that this is not something we can quantitatively evaluate with the current dataset. Hence, we have modified the Title and Abstract to acknowledge the possible involvement of adjacent areas, and also added more discussion about the possible involvement of adjacent regions (fifth paragraph of the Discussion).

While revising the manuscript, we realized that our terminology of “dorsomedial” and “dorsolateral” LH may be somewhat confusing, as the regions are separated more by their dorso-ventral position than by their medio-lateral position (although there is some separation in both dimensions). Hence, we have replaced these terms with dorsolateral and ventrolateral LH (dlLH and vlLH), instead of dmLH and dlLH.

2) In the rabies figure the authors superimpose images from two different mice. This gives the impression that the images come from a single mouse. It is not clear to me how two images from separate mice can accurately be overlayed. I would suggest showing the data as separate images.

We agree that it is hard to assess whether superimposed images were overlaid accurately. Because the RMTg-projecting group had already been shown in Figure 2, its presence in Figure 2—figure supplement 1 (overlaid on top of the VTA-projecting group) was redundant, and hence we simply removed that overlay from the figure supplement.

3) As the authors now split their analysis between early and late responsive cells all the analysis should be done on these two groups. Some of the analysis is still done on the single population, for example the effect of the rEPN inactivation on the baseline firing is still just for the single population. I would suggest splitting the analysis.

We agree that it is important to also analyze the baseline firing in early- versus later-responsive groups. This is now shown in Figure 5C. We found that late- but not early- responsive neurons showed decreased basal activity after the inactivation, and now have added text in results to describe it (Results section “Temporary inactivation of the rEPN reduces LHb basal firings and diminishes LHb responses to motivational stimuli preferentially in late-responsive LHb neurons”). We also applied this split analysis to the percentage of spikes in bursts, and the number of bursts per minute, which also showed inactivation-induced reductions in late- but not early-responders. The only remaining analysis that was not split in this manner was the responses to shocks themselves. This is because the shock cues and shocks are frequently encoded in separate neurons, such that only a few shock cue-responsive neurons also respond to shocks. In other words, most shock-responsive neurons in the LHb are *neither* early *nor* late responsive to cues, as they didn’t respond to cues at all.

4) Cells are reported in the rEPN and LHb that respond to either shocks or cues predicting shocks. Do these ratios change during learning? It would be nice to report if there is a difference early or late in learning as the cue response may increase and the US response decrease over the course of learning as has been reported in primates and mice.

We agree that it is nice to show the acquisition of CS response as the progression of training, like seen in primates and mice. Unfortunately, we did not record rEPN or LHb responses during the training. Animals were already pre-trained with cue-shock association before their first recording session.

Reviewer #2:The authors properly responded to most of my concerns. I have a few further comments.Although the authors showed behavior during reward and neutral trials (Figure 3B), I also would like to see behavior during shock trials.

We now have added behavioral data during reward and shock trials (Figure 3D). It shows that within 2 seconds of cue onset, animals approached the food port in 78% of reward trials but only 19% and 21% of shock and neutral trials, demonstrating accurate discrimination (subsection “Reward cue-inhibited rEPN neurons encode valence”, first paragraph).

Although the authors added Figure 4C showing the effect of rEPN inactivation on food approach behavior, they did not describe anything about this result. They should explain this result in the manuscript, and should discuss why the inactivation did not influence behavior.

We appreciate that the reviewer noted Figure 4C was not explained. We have now added an explanation of this result (subsection “LHb neurons encode motivational valence”, first paragraph). We suspect that the inactivation had no effect on behavior since rEPN inactivation was only done ipsilaterally, leaving the contralateral side completely unaffected.

Reviewer #3:This revised manuscript by Li et al. examined effects of the mouse rostral EP (monkey GPb) on lHb. The original version did not analyze the neuronal phenotypes thoroughly (they heavily depend on single significance tests) and reviewers asked to the authors to examine this in multiple ways, including use of raw firing rates in addition to normalized firing rates, examination of population coding in addition to average, use of different time windows, and examination of RPE or value coding, etc., and to discuss carefully about those factors. In the revision, the authors found that if they divide neurons into two groups, they are able to see some effects on inhibition responses to reward, in addition to excitation responses to shock, in LHb, which was not detected in the original version. It’s good that the authors noticed that you may or may not see effects with different analyses, but I wanted to see more careful and through discussion and analyses.

We agree that thorough analyses are important, but are somewhat confused as to exactly what additional discussion and analyses are being requested. The second sentence of this particular comment suggests the addition of raw firing rates (which we added in the first revision, and further refine in the second revision in Figure 5C), examination of population coding (which was present from the beginning), use of different time windows (which were also present), and examining of RPE vs value coding (which could not be performed because this study did not compare predicted versus unpredicted stimuli). It is not entirely clear to us what else should be added that was not already done.

Just comparing normalized firing rates between pre and post muscimol in different trial types and in arbitrarily divided neurons is misleading. It is not clear how valence coding in LHb changed (or didn’t change, but just scaled down).

We appreciate reviewer’s concerns about how valence coding is changed by rEPN inactivation. Hence, in addition to analysis on inhibited or excited LHb neurons, we also analyzed responses of all recorded LHb neurons to reward, neutral, and shock cues before and after rEPN inactivation (Figure 5L). Prior to rEPN inactivation, population LHb responses demonstrated a monotonically increasing response from reward, to neutral, to shock cues (grey trace in Figure 5L). This was lost after rEPN inactivation (black trace in Figure 5L), suggesting that valence coding in the LHb, is dependent on the rEPN even if we look at all LHb neurons and not just the late-responsive ones.

In Figure 5F and I, here the authors also plotted ratio of post and pre-muscimol firing rates after outcomes, without considering the meaning of coding, and so lost information about inhibition or excitation, compared to baseline or neutral conditions.

We agree that Figures 5F and 5I do not adequately convey the meaning of encoding. We have now reworked the figure to show the post vs pre-muscimol response to cues. We believe that this more directly conveys the EPN effect on the inhibition or excitation from baseline. In addition, information about whether neurons were excited or inhibited by cues before and after the inactivation are shown in Figure 5D, E, G, and H.